# AfriQA: Cross-lingual Open-Retrieval Question Answering for African Languages

Odunayo Ogundepo[1,*,*], Tajuddeen R. Gwadabe[*,*], Clara E. Rivera[2], Jonathan H. Clark[2],
Sebastian Ruder[2], David Ifeoluwa Adelani[3,*], Bonaventure F. P. Dossou[4,5,6,*],
Abdou Aziz DIOP[7,*], Claytone Sikasote[10,*], Gilles Hacheme[9,*], Happy Buzaaba[15,*],
Ignatius Ezeani[14,*], Rooweither Mabuya[16], Salomey Osei[*], Chris Emezue[13,*],
Albert Njoroge Kahira[17,*], Shamsuddeen Hassan Muhammad[18,31,*], Akintunde Oladipo[1,*],
Abraham Toluwase Owodunni[*], Atnafu Lambebo Tonja[12,6,*], Iyanuoluwa Shode[11,*],
Akari Asai[8], Tunde Oluwaseyi Ajayi[19,*], Clemencia Siro[20,*], Steven Arthur[21,*],
Mofetoluwa Adeyemi[1,*], Orevaoghene Ahia[8,*], Anuoluwapo Aremu[*], Oyinkansola Awosan[*],
Chiamaka Chukwuneke[*], Bernard Opoku[22,*], Awokoya Ayodele[23,*], Verrah Otiende[24,*],
Christine Mwase[25,*], Boyd Sinkala[10,*], Andre Niyongabo Rubungo[26,*], Daniel A. Ajisafe[27,*],
Emeka Felix Onwuegbuzia[23,*], Habib Mbow[28,*], Emile Niyomutabazi[29,*], Eunice Mukonde[10,*],
Falalu Ibrahim Lawan[30,*], Ibrahim Said Ahmad[31,*], Jesujoba O. Alabi[32,*],
Martin Namukombo[33,*], Mbonu Chinedu[35,*], Mofya Phiri[10,*], Neo Putini[25,*],
Ndumiso Mngoma[31,*], Priscilla A. Amuok[*], Ruqayya Nasir Iro[32,*], Sonia Adhiambo[34,*]

[*]Masakhane NLP, [1]University of Waterloo, Canada, [2]Google Research, [3]University College London,
[4]Mila Quebec AI Institute, [5]McGill University, [6]Lelapa AI, [7]GalsenAI,
[8]University of Washington, [9]Ai4Innov, [10] University of Zambia, [11]Montclair State University,
[12]Instituto Politécnico Nacional, Mexico, [13]Technical University of Munich, [14]Lancaster University,
[15]RIKEN Center for AIP, [16]South African Centre for Digital Language Resources, [17]Jülich Supercomputing Centre,
[18]University of Porto, [19]Insight Centre for Data Analytics, [20]University of Amsterdam, [21]Accra Institute of Technology,
[22]Kwame Nkrumah University of Science and Technology, [23]University of Ibadan, [24]Tom Mboya University
[25]Fudan University, [26]University of Electronic Science and Technology of China, [27]The University of British Columbia,
[28]African Master in Machine Intelligence, [29]College de Rebero, [30]Kaduna State University, [31]Bayero University Kano
[32]Saarland University, Germany, [33]University of Edinburgh, [34] Kenyatta University, [35]Nnamdi Azikiwe University

## Abstract

African languages have far less in-language content available digitally, making it challenging for question-answering systems to satisfy the information needs of users. Cross-lingual open-retrieval question answering (XOR QA) systems—those that retrieve answer content from other languages while serving people in their native language—offer a means of filling this gap. To this end, we create AFRIQA, the first cross-lingual QA dataset with a focus on African languages. AFRIQA includes 12,000+ XOR QA examples across 10 African languages. While previous datasets have focused primarily on languages where cross-lingual QA *augments* coverage from the target language, AFRIQA focuses on languages where cross-lingual answer content is the *only* high-coverage source of answer content. Because of this, we argue that African languages are one of the most important and realistic use cases for XOR QA. Our experiments demonstrate the poor performance of automatic trans-

lation and multilingual retrieval methods. Overall, AFRIQA proves challenging for state-of-the-art QA models. We hope that the dataset enables the development of more equitable QA technology. [1]

## 1 Introduction

Question Answering (QA) systems provide access to information (Kwiatkowski et al., 2019) and increase accessibility in a range of domains, from healthcare and health emergencies such as COVID-19 (Möller et al., 2020; Morales et al., 2021) to legal queries (Martinez-Gil, 2021) and financial questions (Chen et al., 2021). Many of these applications are particularly important in regions where information and services may be less accessible and where language technology may thus help to reduce the burden on the existing system. At the same time, many people prefer to access information in their local languages—or simply do not speak a

---

[*]Equal contribution.

[1]The data is available at:
https://github.com/masakhane-io/afriqa and
https://huggingface.co/datasets/masakhane/afriqa

| Dataset | QA? | CLIR? | Open Retrieval? | # Languages | # African Languages |
|---|---|---|---|---|---|
| XQA (Liu et al., 2019) | ✓ | ✓ | ✓ | 9 | Nil |
| XOR QA (Asai et al., 2021) | ✓ | ✓ | ✓ | 7 | Nil |
| XQuAD (Artetxe et al., 2020) | ✓ | ✗ | ✗ | 11 | Nil |
| MLQA (Lewis et al., 2020) | ✓ | ✗ | ✗ | 7 | Nil |
| MKQA (Longpre et al., 2021) | ✓ | ✗ | ✓ | 26 | Nil |
| TyDi QA (Clark et al., 2020) | ✓ | ✗ | ✓ | 11 | 1 |
| AmQA (Abedissa et al., 2023) | ✓ | ✗ | ✗ | 1 | 1 |
| KenSwQuAD (Wanjawa et al., 2023) | ✓ | ✗ | ✗ | 1 | 1 |
| AFRIQA (Ours) | ✓ | ✓ | ✓ | 10 | 10 (see Table 3) |

Table 1: **Comparison of the Dataset with Other Question Answering Datasets.** This table provides a comparison of the current dataset used in the study with other related datasets. The first, second, and third columns, "QA", "CLIR", and "Open Retrieval", indicate whether the dataset is question answering, cross-lingual or open retrieval, respectively. The fourth column, "# Languages", shows the total number of languages in the dataset. The final column lists the African languages present in the dataset.

language supported by current language technologies (Amano et al., 2016). To benefit the more than three billion speakers of under-represented languages around the world, it is thus crucial to enable the development of QA technology in local languages.

Standard QA datasets mainly focus on English (Joshi et al., 2017; Mihaylov et al., 2018; Kwiatkowski et al., 2019; Sap et al., 2020). While some reading comprehension datasets are available in other high-resource languages (Ruder and Sil, 2021), only a few QA datasets (Clark et al., 2020; Asai et al., 2021; Longpre et al., 2021) cover a typologically diverse set of languages—and very few datasets include African languages (see Table 1).

In this work, we lay the foundation for research on QA systems for one of the most linguistically diverse regions by creating AFRIQA, the first QA dataset for 10 African languages. AFRIQA focuses on open-retrieval QA where information-seeking questions[2] are paired with retrieved documents in which annotators identify an answer if one is available (Kwiatkowski et al., 2019). As many African languages lack high-quality in-language content online, AFRIQA employs a cross-lingual setting (Asai et al., 2021) where relevant passages are retrieved in a high-resource language spoken in the corresponding region and answers are translated into the source language. To ensure utility of this dataset, we carefully select a relevant source

language (either English or French) based on its prevalence in the region corresponding to the query language. AFRIQA includes 12,000+ examples across 10 languages spoken in different parts of Africa. The majority of the dataset's questions are centered around entities and topics that are closely linked to Africa. This is an advantage over simply translating existing datasets into these languages. By building a dataset from the ground up that is specifically tailored to African languages and their corresponding cultures, we are able to ensure better contextual relevance and usefulness of this dataset.

We conduct baseline experiments for each part of the open-retrieval QA pipeline using different translation systems, retrieval models, and multilingual reader models. We demonstrate that cross-lingual retrieval still has a large deficit compared to automatic translation and retrieval; we also show that a hybrid approach of sparse and dense retrieval improves over either technique in isolation. We highlight interesting aspects of the data and discuss annotation challenges that may inform future annotation efforts for QA. Overall, AFRIQA proves challenging for state-of-the-art QA models. We hope that AFRIQA encourages and enables the development and evaluation of more multilingual and equitable QA technology. The dataset will be released under the Creative Commons Attribution 4.0 International (CC BY 4.0) license.

In summary, we make the following contributions:

- We introduce the first cross-lingual question answering dataset with 12,000+ questions across 10 geographically diverse African languages. This dataset directly addresses

---

[2]These questions are **information-seeking** in that they are written without seeing the answer, as is the case with real users of QA systems. We contrast this with the reading comprehension task where the question-writer sees the answer passage prior to writing the question; this genre of questions tends to have both higher lexical overlap with the question and elicit questions that may not be of broad interest.

| lang | Question $Q_L$ (*Translation $Q_{pl}$*) | Relevant Passage $P_{pl}$ | Answer $A_{pl}$ (*Translation $A_L$*) |
|---|---|---|---|
| hau | Jahohi nawa ne a kasar Malaysia? banga? (*How many states are there in Malaysia?*) | The states and federal territories of Malaysia are the principal administrative divisions of Malaysia. Malaysia is a federation of **13** states (Negeri) and 3 federal territories. | 13 (*13*) |
| bem | Bushe Mwanawasa stadium ingisha abantu banga? (*What is the capacity of Mwanawasa Stadium?*) | The Levy Mwanawasa Stadium is a multi-purpose stadium in Ndola, Zambia. It is used mostly for football matches. The stadium has a capacity of **49,800 people**. | 49,800 people (*Abantu 49800*) |
| wol | Man po moo niroo ag powum Softbal? (*Quel sport ressemble beaucoup au softball?*) | Ce sport est un descendant direct du **baseball** (afin de différencier les deux) mais diffère de ce dernier par différents aspects dont les cinq principaux sont les suivants. | baseball (*Bas-bal*) |

Table 2: Table showing selected questions, relevant passages, and answers in different languages from the dataset. It also includes the human-translated versions of both questions and answers. For the primary XOR QA task, systems are expected to find the relevant passage among all Wikipedia passages, not simply the gold passage shown above.

the deficit of African languages in existing datasets.

- We conduct an analysis of the linguistic properties of the 10 languages, which is crucial to take into account when formulating questions in these languages.

- Finally, we conduct a comprehensive evaluation of the dataset for each part of the open-retrieval QA pipeline using various translation systems, retrieval models, and multilingual reader models.

## 2 AFRIQA

AFRIQA is a cross-lingual QA dataset that was created to promote the representation and coverage of under-resourced African languages in NLP research. We show examples of the data in Table 2. In §2.1, we provide an overview of the 10 languages discussing their linguistic properties, while §2.2 and §2.3 describe the data collection procedure and quality control measures put in place to ensure the quality of the dataset.

### 2.1 Discussion of Languages

African languages have unique typologies, grammatical structures, and phonology, many of them being tonal and morphologically rich (Adelani et al., 2022b). We provide a high-level overview of the linguistic properties of the ten languages in AFRIQA that are essential to consider when crafting questions for QA systems.

**Bemba**, a morphologically rich Bantu language, uses affixes to alter grammatical forms. Com-

mon question words in Bemba include "cinshi" (what), "naani"(who), "liisa" (when), "mulandun-shi" (why), "ciisa" (which), "kwi/kwiisa" (where), and "shaani" (how).

**Fon** is an isolating language in terms of morphology typology. Common question wh-words in Fon are Etέ(what), Mɛ (who), Hwetɛ́nu (when), Aniwú (why), ɖe tɛ (which) and Fitɛ (where).

**Hausa** is the only Afro-Asiatic language in AFRIQA. It typically makes use of indicative words for changes to the grammatical forms within a sentence, such as negation, tenses, and plurality. For example, "hula" (cap) – "huluna" (caps), "mace" (girl) – "mataye" (girls). Typical question wh-words used are "me/ya" (what), "wa"(who), "yaushe" (when), "dan me/akan me" (why), "wanne" (which), "ina/ a ina" (where), and "yaya/qaqa" (how).

**Igbo** is a morphologically rich language and most changes in grammatical forms (negations, questions) can be embedded in a single word or by varying the tone. Question words are often preceded by "kedu" or "gini" like "kedu/gini" (what), "onye/kedu onye" (who), "kedu mgbe" (when), "gini mere/gini kpatara" (why), "kedu nke" (which), "ebee" (where), and "kedu ka" or "kedu etu" (how).

**Kinyarwanda** is a morphologically rich language with several grammatical features such as negation, tenses, and plurals that are expressed as changes to morphemes in a word. Question words typically used are "iki" (what), "nde/inde" (who), 'ryari'" (when), "ikihe/uwuhe" (which), "hehe" (where), and "gute" (how).

**Swahili** is a morphologically rich language that

typically has several morphemes to incorporate changes to grammatical forms such as negation, tenses and plurality. A question word can be placed at the beginning or end of the question sentence, for example, "amekuja nani?" (who has come?) and "nani amekuja?" (who has come?). Other question words often used are "nini" (what), "nani" (who), "lini" (when), "kwanini" (why), "wapi", (where), and "vipi" (how).

**Twi** is a dialect of the Akan language and AFRIQA includes the Asante variant. A few common question words used in Twi are "ɛdeɛn"(what), "hwan" (who), "daben" (when), "adɛn" (why), "deɛhen" (which), "ɛhenfa" (where), "sɛn" (how).

**Wolof** is an agglutinative language, and unlike other Bantu languages, it utilizes dependent words rather than affixes attached to the headwords for grammatical modifications. Common question words in Wolof include "ian" (what), "kan" (who), "kañ" (when), "lu tax", "ban" (which), "fan" (where), and "naka" (how).

**Yorùbá** has a derivational morphology that entails affixation, reduplication, and compounding. Yorùbá employs polar question words such as "nje", "se", "abi", "sebi" (for English question words "do" or "is", "are", "was" or "were") and content question markers such as "tani" (who), "kini" (what), "nibo" (where), "elo/meloo" (how many), "bawo"(how is), "kilode" (why), and "igba/nigba" (when). Negation can be expressed with "kò".

**Zulu** is a very morphologically-rich language where several grammatical features such as tense, negation, and the plurality of words are indicated through prefixes or suffixes. The most commonly used question words in Zulu are "yini" (what), "ubani" (who), "nini" (when), "kungani" (why), "yiliphi" (which), "kuphi" (where), "kanjani" (how), and "yenza" (do).

## 2.2 Data Collection Procedure

For each of the 10 languages in AFRIQA, a team of 2–6 native speakers was responsible for the data collection and annotation. Each team was led by a coordinator. The annotation pipeline consisted of 4 distinct stages: 1) question elicitation in an African language; 2) translation of questions into a pivot language; 3) answer labeling in the pivot language based on a set of candidate paragraphs; and 4) answer translation back to the source language. All data contributions were compensated financially.

### 2.2.1 Question Elicitation

The TyDi QA methodology (Clark et al., 2020) was followed to elicit locally relevant questions. Team members were presented with prompts including the first 250 characters of the most popular Wikipedia[3] articles in their languages, and asked to write factual or procedural questions for which the answers were not contained in the prompts. Annotators were encouraged to follow their natural curiosity. This annotation process avoids excessive and artificial overlap between the question and answer passage, which can often arise in data collection efforts for non-information-seeking QA tasks such as reading comprehension.[4] For languages like Fon and Bemba without a dedicated Wikipedia, relevant prompts from French and English Wikipedia were used to stimulate native language question generation. For Swahili, unanswered TyDi QA questions were curated for correctness. The inability of the original TyDi QA team to locate a suitable Swahili paragraph to answer these questions necessitated their inclusion. Simple spreadsheets facilitated this question elicitation process.

Before moving on to the second stage, team coordinators reviewed elicited questions for grammatical correctness and suitability for the purposes of information-seeking QA.

### 2.2.2 Question Translation

Elicited questions were translated from the original African languages into pivot languages following Asai et al. (2021). English was used as the pivot language across all languages except Wolof and Fon, for which French was used.[5] Where possible, questions elicited by one team member were allocated to a different team member for translation to further ensure that only factual or procedural questions that are grammatically correct make it into the final dataset. This serves as an additional validation layer for the elicited questions.

---

[3] https://www.wikipedia.org/
[4] Reading comprehension differs from information-seeking QA as question-writers see the answer prior to writing the question and thus tests understanding of the answer text rather than the general ability to provide a correct answer.
[5] French is widely used in the regions where Fon and Wolof are spoken, so there may be a higher probability of finding answers in French than in other pivot languages.

### 2.2.3 Answer Retrieval

Using the translated questions as queries, Google Programmable Search Engine [6] was used to retrieve Wikipedia paragraphs that are candidates to contain an answer in the corresponding pivot language. The Mechanical Turk interface was employed—all annotations were carried out by team members. was used to show candidate paragraphs to team members who were then asked to identify 1) the paragraph that contains an answer and 2) the exact minimal span of the answer. In the case of polar questions, team members had to select "Yes" or "No" instead of the minimal span. In cases where candidate paragraphs did not contain the answer to the corresponding question, team members were instructed to select the "No gold paragraph" option.

As with question elicitation, team members went through a phase of training, which included a group meeting where guidelines were shared and annotators were walked through the labeling tool. Two rounds of in-tool labeling training were conducted.

### 2.2.4 Answer Translation

To obtain answers in the African languages, we translated the answers in the pivot languages to the corresponding African languages. We allocated the task of translating the answers labeled by team members to different team members in order to ensure accuracy. Translators were instructed to minimize the span of the translated answers. In cases where the selected answers were incorrect or annotators failed to select the minimum span, we either removed the question, corrected the answer, or re-annotated the question using the annotation tool.

### 2.3 Quality Control

We enforced rigorous quality control measures throughout the dataset generation process to ascertain its integrity, quality, and appropriateness. Our strategy involved recruiting native language speakers as annotators and team coordinators. Each annotator underwent initial training in question elicitation via English prompts, with personalized feedback focusing on factual question generation and avoiding prompt-contained answers. Annotators were required to achieve at least 90% accuracy to proceed to native language elicitation, with additional one-on-one training rounds provided as necessary.

Each language team comprised a minimum of three members, with Fon and Kinyarwanda teams as exceptions, hosting two members each. This structure was designed to ensure different team members handled question elicitation and translation, enhancing quality control. Non-factual or inappropriate questions were flagged during the translation and answer labeling phases, leading to their correction or removal. Team coordinators meticulously reviewed all question-and-answer pairs alongside their translations, while central managers checked translation consistency. Post-annotation controls helped rectify common issues such as answer-span length and incorrect answer selection.

### 2.4 Final Dataset

The statistics of the dataset are presented in Table 3, which includes information on the languages, their corresponding pivot languages, and the total number of questions collected for each language. The final dataset consists of a total of 12,239 questions across 10 different languages, with 8,892 corresponding question-answer pairs. We observed a high answer coverage rate, with only 27% of the total questions being unanswerable using Wikipedia. This can be attributed to the lack of relevant information on Wikipedia, especially for Africa related entities with sparse information. Despite this sparsity, we were able to find answers for over 60% of the questions in most of the languages in our collection.

## 3 Tasks and Baselines

As part of the evaluation for AFRIQA, we follow the methodology proposed in Asai et al. (2021) and assess its performance on three different tasks: XOR-Retrieve, XOR-PivotLanguageSpan, and XOR-Full. Each task poses unique challenges for cross-lingual information retrieval and QA due to the low-resource nature of many African languages.

## 4 Experiments

### 4.1 Translation Systems

A common approach to cross-lingual QA is to translate queries from the source language into a target language, which is then used to find an answer in a given passage. For our experiments, we explore the use of different translation systems as baselines for AFRIQA. We consider human translation, Google

---

[6] https://developers.google.com/custom-search/

| Source Language | ISO | Pivot Language | African Region | Script | # Native Speakers | Train | Dev | Test | % Unanswerable Questions |
|---|---|---|---|---|---|---|---|---|---|
| Bemba | bem | English | South, East & Central | Latin | 4M | 502 | 503 | 314 | 0.41 |
| Fon | fon | French | West | Latin | 2M | 427 | 428 | 386 | 0.22 |
| Hausa | hau | English | West | Latin | 63M | 435 | 436 | 300 | 0.36 |
| Igbo | ibo | English | West | Latin | 27M | 417 | 418 | 409 | 0.18 |
| Kinyarwanda | kin | English | Central | Latin | 15M | 407 | 409 | 347 | 0.26 |
| Swahili | swa | English | East & Central | Latin | 98M | 415 | 417 | 302 | 0.34 |
| Twi | twi | English | West | Latin | 9M | 451 | 452 | 490 | 0.12 |
| Wolof | wol | French | West | Latin | 5M | 503 | 504 | 334 | 0.38 |
| Yorùbá | yor | English | West | Latin | 42M | 360 | 361 | 332 | 0.21 |
| Zulu | zul | English | South | Latin | 27M | 387 | 388 | 325 | 0.26 |
| Total | — | — | — | — | 292M | 4333 | 4346 | 3560 | 0.27 |

Table 3: **Dataset information:** This table contains key information about the AFRIQA Dataset

Translate, and open-source translation models such as NLLB (NLLB Team et al., 2022) and finetuned M2M-100 models (Adelani et al., 2022a) in zero-shot settings.

**Google Translate.** We use Google Translate because it provides out-of-the-box translation for 7 out of 10 languages in our dataset. Although Google Translate provides a strong translation baseline for many of the languages, we cannot guarantee the future reproducibility of these translations as it is a product API and is constantly being updated. For our experiments, we use the translation system as of February 2023 [7].

**NLLB.** NLLB is an open-source translation system trained on 100+ languages and provides translation for all the languages in AFRIQA. At the time of release, NLLB provides state–of–the–art translation in many languages and covers all the languages in our dataset. For our experiments, we use the 1.3B size NLLB models.

### 4.2 Passage Retrieval (XOR-Retrieve)

We present two baseline retrieval systems: translate–retrieve and cross-lingual baselines. In the translate–retrieve baseline, we first translate the queries using the translation systems described in §4.1. The translated queries are used to retrieve relevant passages using different retrieval systems outlined below. Alternatively, the cross-lingual baseline directly retrieves passages in the pivot language without the need for translation using a multilingual dense retriever.

**BM25.** BM25 (Robertson and Zaragoza, 2009)

is a classic term-frequency-based retrieval model that matches queries to relevant passages using the frequency of word occurrences in both queries and passages. We use the BM25 implementation provided by Pyserini (Lin et al., 2021) with default hyperparameters k1 = 0.9, b = 0.4 for all languages.

**mDPR.** We evaluate the performance of mDPR, a multilingual adaptation of the Dense Passage Retriever (DPR) model (Karpukhin et al., 2020) using multilingual BERT (mBERT). We finetuned mDPR on the MS MARCO passage ranking dataset (Bajaj et al., 2018) for our experiments. Retrieval is performed using the Faiss Flat Index implementation provided by Pyserini.

**Sparse–Dense Hybrid.** We also explore sparse–dense hybrid baselines, a combination of sparse (BM25) and hybrid (mDPR) retrievers. We use a linear combination of both systems to generate a reranked list of passages for each question.

### 4.3 Answer Span Prediction

To benchmark models' answer selection capabilities on AFRIQA, we combine different translation, extractive, and generative QA approaches.

**Generative QA on Gold Passages.** To evaluate the performance of generative QA, we utilize mT5–base (Xue et al., 2021) finetuned on SQuAD 2.0 (Rajpurkar et al., 2016) and evaluate it using both translated and original queries. The model was provided with the queries and the gold passages that were annotated using a template prompt and generates the answers to the questions.

**Extractive QA on Retrieved Passages.** For XOR-PivotLanguageSpan baselines, we employed an extractive QA model that extracts the answer span from the retrieved passages produced by the

---

[7]Note that while Google Translate supports 133 languages, it does not include Bemba, Fon, nor Wolof.

| lang | Human Translation | | | GMT | | NLLB | | M2M-100 | | Crosslingual |
| | BM25 | mDPR | Hybrid | BM25 | mDPR | BM25 | mDPR | BM25 | mDPR | mDPR |
|---|---|---|---|---|---|---|---|---|---|---|
| | | | | | Recall@10 | | | | | |
| bem | 55.7 | 67.5 | **72.3** | — | — | 52.2 | **59.8** | — | — | 14.7 |
| fon | 66.3 | 69.4 | **70.7** | — | — | 43.9 | **48.7** | 39.9 | 43.3 | 28.5 |
| hau | 58.0 | 65.7 | **72.7** | 53.3 | **60.3** | 52.0 | 59.7 | 36.7 | 44.3 | 13.7 |
| igb | 70.4 | 74.3 | **82.9** | 65.5 | **71.2** | 64.8 | 68.0 | 62.1 | 67.5 | 25.4 |
| kin | 59.1 | 66.3 | **75.5** | 53.6 | **61.1** | 53.0 | 58.8 | — | — | 15.6 |
| swa | 46.0 | 61.9 | **67.6** | 45.0 | **60.9** | 43.1 | 58.3 | 39.1 | 54.6 | 20.9 |
| twi | 61.8 | 66.7 | **75.3** | 56.1 | **58.0** | 50.4 | 54.1 | 45.7 | 49.4 | 21.4 |
| wol | 61.4 | 67.7 | **68.6** | — | — | 35.0 | **36.5** | 34.4 | 35.0 | 13.8 |
| yor | 55.1 | 66.6 | **71.7** | 52.1 | **59.0** | 50.9 | 57.5 | 36.8 | 35.5 | 21.4 |
| zul | 59.7 | 70.2 | **76.3** | 57.2 | **66.2** | 51.5 | 64.6 | 45.5 | 60.0 | 14.2 |
| avg | 59.4 | 67.6 | **73.4** | 54.7 | **62.4** | 49.7 | 56.6 | 42.5 | 48.7 | 19.0 |

Table 4: **Retrieval Recall@10**: This table displays the retrieval recall results for various translation types on the test set of AFRIQA. The table shows the percentage of retrieved passages that contain the answer for the top-10 retrieved passages. The last column represents crosslingual retrieval, where we skip the translation step and use the original queries. We boldface the best-performing model for each language within the human translation oracle scenario and within the real-world automatic translation scenario.

| | HT | | GMT | | NLLB | | Crosslingual | |
| | F1 | EM | F1 | EM | F1 | EM | F1 | EM |
|---|---|---|---|---|---|---|---|---|
| bem | **48.8** | **41.7** | — | — | 38.5 | 32.0 | 2.9 | 1.1 |
| fon | **41.4** | **28.5** | — | — | 23.4 | 15.3 | 5.1 | 2.3 |
| hau | **58.5** | **49.0** | 53.5 | 45.7 | 50.9 | 42.7 | 25.8 | 22.3 |
| ibo | **66.6** | **59.2** | 59.8 | 53.3 | 60.2 | 53.3 | 41.7 | 34.7 |
| kin | **60.8** | **43.8** | 57.3 | 40.9 | 58.8 | 42.9 | 25.5 | 20.2 |
| swa | **52.3** | **42.6** | 48.9 | 40.8 | 49.2 | 41.2 | 29.4 | 23.5 |
| twi | **55.4** | **45.3** | 42.0 | 33.7 | 40.1 | 33.1 | 5.3 | 3.5 |
| wol | **44.6** | **36.1** | — | — | 21.8 | 16.9 | 3.9 | 2.8 |
| yor | **54.9** | **49.8** | 48.9 | 45.1 | 47.9 | 43.0 | 11.9 | 7.8 |
| zul | **60.2** | **50.8** | 57.4 | 48.9 | 55.6 | 46.5 | 24.7 | 20.9 |
| avg | **54.5** | **44.7** | 46.0 | 38.6 | 44.6 | 36.7 | 17.6 | 13.9 |

Table 5: **Generative Gold Passages Answer Prediction:** Comparison of F1 and Exact Match Accuracy scores for generative answer span prediction on the test set using mT5-base (Xue et al., 2020) as the backbone.

| | HT | | GMT | | NLLB | | Crosslingual | |
| | F1 | EM | F1 | EM | F1 | EM | F1 | EM |
|---|---|---|---|---|---|---|---|---|
| bem | **38.2** | **29.5** | — | — | 30.0 | 21.9 | 0.4 | 0.4 |
| fon | **53.8** | **40.4** | — | — | 37.5 | 26.7 | 13.4 | 6.0 |
| hau | **60.9** | **52.7** | 54.4 | 47.7 | 50.9 | 43.7 | 27.7 | 23.7 |
| ibo | **68.2** | **60.6** | 62.1 | 55.0 | 62.8 | 56.2 | 29.2 | 24.7 |
| kin | **56.8** | **38.9** | 50.8 | 36.0 | 51.3 | 36.6 | 22.7 | 17.9 |
| swa | **45.2** | **37.9** | 44.6 | 37.9 | 45.2 | 38.1 | 31.6 | 24.6 |
| twi | **51.2** | **41.8** | 39.2 | 31.1 | 34.3 | 30.0 | 3.4 | 2.5 |
| wol | **45.2** | **33.9** | — | — | 33.2 | 26.0 | 1.8 | 0.9 |
| yor | **45.1** | **38.6** | 36.0 | 31.7 | 32.3 | 28.0 | 6.0 | 3.8 |
| zul | **59.1** | **49.2** | 56.0 | 48.6 | 53.6 | 45.8 | 17.0 | 13.5 |
| avg | **52.4** | **42.4** | 42.9 | 36.0 | 43.1 | 35.3 | 15.3 | 11.8 |

Table 6: **Extractive Gold Passages Answer Prediction:** Comparison of F1 and Exact Match Accuracy scores for extractive answer span prediction on the test set using AfroXLMR-base (Alabi et al., 2022) as the backbone.

various retrieval baselines outlined in §4.2. The model is trained to extract answer spans from each passage, along with the probability indicating the likelihood of each answer. The answer span with the highest probability is selected as the correct answer. We trained a multilingual DPR reader model, which was initialized from mBERT and finetuned on Natural Questions (Kwiatkowski et al., 2019).

## 5  Results and Analysis

### 5.1  XOR-Retrieve Results

We present the retrieval results for recall@10 in Table 4 [8]. The table includes retriever results

using different translation and retrieval systems. We also report the performance with both original and human-translated queries. The table shows that hybrid retrieval using human translation yields the best results for all languages, with an average recall@10 of 73.9. In isolation, mDPR retrieval outperforms BM25 for all translation types. This table also enables us to compare the effectiveness of different translation systems in locating relevant passages for cross-lingual QA in African languages. This is illustrated in Figure 1, showing retriever recall rates for different translation types at various

---

[8] For recall@k retrieval results, we assume that there is only one gold passage despite the possibility of other retrieved passages containing the answer.

| Query Translation | Retrieval | Pivot Language Span F1 |||||||||| Average ||
|---|---|---|---|---|---|---|---|---|---|---|---|---|---|
| | | bem | fon | hau | ibo | kin | swa | twi | wol | yor | zul | F1 | EM |
| HT | BM25 | 29.2 | **11.4** | 31.4 | 43.0 | 33.8 | 24.3 | 38.4 | 15.4 | 28.9 | 32.8 | 28.9 | 19.9 |
| HT | mDPR | 32.5 | 11.0 | **35.8** | 44.8 | 35.4 | 28.2 | 40.7 | 14.7 | 31.7 | **36.5** | 31.1 | 21.5 |
| HT | Hybrid | **34.7** | 11.3 | 35.5 | **46.1** | **39.2** | 27.5 | **41.8** | **16.2** | **32.4** | 34.6 | **32.0** | **21.9** |
| GMT | BM25 | — | — | 21.0 | 38.6 | 28.3 | 24.7 | 27.7 | — | 21.7 | 31.6 | 27.7 | 21.2 |
| GMT | mDPR | — | — | 31.5 | 39.3 | 35.3 | **29.1** | 31.1 | — | 22.9 | 36.0 | 32.2 | 22.3 |
| NLLB | BM25 | 23.8 | 3.6 | 24.6 | 37.6 | 29.3 | 25.2 | 25.7 | 4.4 | 17.3 | 26.8 | 19.8 | 13.8 |
| NLLB | mDPR | 24.1 | 5.1 | 27.2 | 39.6 | 33.3 | 25.9 | 28.2 | 5.2 | 21.4 | 30.4 | 24.0 | 16.0 |

Table 7: F1 scores on pivot language answer generation using an extractive multilingual reader model with different query translation and retrieval methods.

| Translation || | XOR-Full F1 |||||||||| Average |||
|---|---|---|---|---|---|---|---|---|---|---|---|---|---|---|---|
| Query | Answer | Retrieval | bem | fon | hau | ibo | kin | swa | twi | wol | yor | zul | F1 | EM | BLEU |
| GMT | GMT | BM25 | — | — | 20.4 | 30.4 | 24.2 | 18.1 | 14.9 | — | 16.1 | 19.7 | 20.5 | 12.1 | 18.3 |
| GMT | GMT | mDPR | — | — | **21.7** | **33.0** | **26.5** | **21.9** | 16.5 | 14.2 | **20.4** | **21.1** | **23.0** | **14.2** | **20.7** |
| NLLB | NLLB | BM25 | **13.6** | 2.6 | 17.5 | 26.5 | 19.9 | 19.2 | 18.4 | 3.2 | 12.7 | 12.5 | 14.6 | 7.5 | 12.9 |
| NLLB | NLLB | mDPR | 13.3 | 4.3 | 19.3 | 29.9 | 22.4 | 20.3 | **19.5** | **3.5** | 17.6 | 13.1 | 16.3 | 8.3 | 14.3 |

Table 8: XOR-Full F1 results combining different translation and retriever systems.

cutoffs using mDPR.

We observe that human translation yields better accuracy than all other translation types, indicating that the current state-of-the-art machine translation systems still have a long way to go in accurately translating African languages. Google Translate shows better results for the languages where it is available, while the NLLB model provides better coverage. The cross-lingual retrieval model that retrieves passages using questions in their original language is the least effective of all the model types. This illustrates that the cross-lingual representations learned by current retrieval methods are not yet of sufficient quality to enable accurate retrieval across different languages.

### 5.2 XOR-PivotLanguageSpan Results

**Gold Passage Answer Prediction.** We first evaluate the generative QA setting using gold passages. We present F1 and Exact Match results using different methods to translate the query in Table 5. Human translation of the queries consistently outperforms using machine-translated queries, which outperforms using queries in their original language.

**Retrieved Passages Answer Prediction.** We now evaluate performance using retrieved passages from §5.1. We present F1 and Exact Match results with different translation–retriever combinations in Table 7. We extract the answer spans from only the

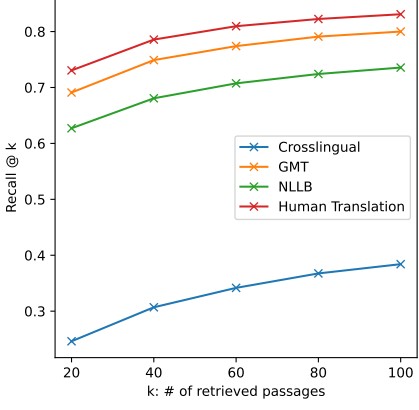

Figure 1: Graph of retriever recall@k for different translation systems. The scores shown in this graph are from mDPR retrieval.

top-10 retrieved passages for each question using an extractive multilingual reader model (see §4.3). The model assigns a probability to each answer span, and we select the answer with the highest probability as the final answer.

Our results show that hybrid retrieval using human-translated queries achieves the best performance across all languages on average. Using human-translated queries generally outperforms using translations by both Google Translate and NLLB, regardless of the retriever system used. In terms of retrieval methods, mDPR generally performs better than BM25, with an average gain of 3 F1 points across different translation types. These

results highlight the importance of carefully selecting translation–retriever combinations to achieve the best answer span prediction results in cross-lingual QA.

## 5.3 XOR-Full Results

Each pipeline consists of components for question translation, passage retrieval, answer extraction, and answer translation. From Table 8, we observe that Google machine translation combined with mDPR is the most effective. This is followed by a pipeline combining NLLB translation with mDPR.

## 6 Related Work

**Africa NLP.** In parallel with efforts to include more low-resource languages in NLP research (Costa-jussà et al., 2022; Ruder, 2020), demand for NLP that targets African languages, which represent more than 30% of the world's spoken languages (Ogueji et al., 2021) is growing. This has resulted in the creation of publicly available multilingual datasets targeting African languages for a variety of NLP tasks such as sentiment analysis (Muhammad et al., 2023; Shode et al., 2022), language identification (Adebara et al., 2022), data-to-text generation (Gehrmann et al., 2022), topic classification (Adelani et al., 2023; Hedderich et al., 2020), machine translation (Adelani et al., 2022a; Nekoto et al., 2020), and NER (Eiselen, 2016; Adelani et al., 2021, 2022b).

Datasets for QA and Information Retrieval tasks have also been created. They are, however, very few and cater to individual languages (Abedissa et al., 2023; Wanjawa et al., 2023) or a small subset of languages spoken in individual countries (Daniel et al., 2019; Zhang et al., 2022). Given the region's large number of linguistically diverse and information-scarce languages, multilingual and cross-lingual datasets are encouraged to catalyze research efforts. To the best of our knowledge, there are no publicly available cross-lingual open-retrieval African language QA datasets.

**Comparison to Other Resources.** Multilingual QA datasets have paved the way for language models to simultaneously learn across multiple languages, with both reading comprehension (Lewis et al., 2020) and other QA datasets (Longpre et al., 2021; Clark et al., 2020) predominantly utilizing publicly available data sources such as Wikipedia, SQUAD, and the Natural Questions dataset. To address the information scarcity of the typically used data sources for low-resource languages, cross-lingual datasets (Liu et al., 2019; Asai et al., 2021) emerged that translate between low-resource and high-resource languages, thus providing access to a larger information retrieval pool which decreases the fraction of unanswerable questions. Despite these efforts, however, the inclusion of African languages remains extremely rare, as shown in Table 1, which compares our dataset to other closely related QA datasets. TyDi QA features Swahili as the sole African language out of the 11 languages it covers.

## 7 Conclusion

In this work, we take a step toward bridging the information gap between native speakers of many African languages and the vast amount of digital information available on the web by creating AFRIQA, the first open-retrieval cross-lingual QA dataset focused on African languages with 12,000+ questions. We anticipate that AFRIQA will help improve access to relevant information for speakers of African languages. By leveraging the power of cross-lingual QA, we hope to bridge the information gap and promote linguistic diversity and inclusivity in digital information access.

## Limitations

Our research focuses on using English and French Wikipedia as the knowledge base for creating systems that can answer questions in 10 African languages. While Wikipedia is a comprehensive source of knowledge, it does not accurately reflect all societies and cultures (Callahan and Herring, 2011). There is a limited understanding of African contexts with relatively few Wikipedia articles dedicated to Africa related content. This could potentially limit the ability of a QA system to accurately find answers to questions related to African traditions, practices, or entities. Also, by focusing on English and French and pivot languages, we might introduce some translation inaccuracies or ambiguities which might impact the performance of a QA system.

Addressing these limitations requires concerted efforts to develop more localized knowledge bases or improve existing sources such as Wikipedia by updating or creating articles that reflect the diversity and richness of African societies.

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

| Parameters | Value |
|---|---|
| backbone | multilingual-bert |
| # train epochs | 25 |
| # warmup steps | 500 |
| # GPUs | 4 |
| # gradient accumulation | 2 |
| learning rate | 5.0e-05 |
| $\epsilon$ | 1.0e-08 |
| batch size | 16 |
| weight decay | 0.01 |
| max gradient norm | 1.0 |
| seed | 42 |
| max sequence length | 256 |

Table 9: **DPR Reader Training Configurations**

Xinyu Zhang, Nandan Thakur, Odunayo Ogundepo, Ehsan Kamalloo, David Alfonso-Hermelo, Xiaoguang Li, Qun Liu, Mehdi Rezagholizadeh, and Jimmy Lin. 2022. Making a miracl: Multilingual information retrieval across a continuum of languages.

## A  Preparing Wikipedia Passages

Wikipedia is a popular choice as a knowledge base for open-retrieval question-answering (QA) experiments, where articles are usually divided into fixed-length passages that are indexed and used for retrieval and reading comprehension, as seen in previous works such as (Karpukhin et al., 2020; Asai et al., 2021). However, Tamber et al. (2023) highlighted that splitting articles into fragmented and disjoint passages can negatively impact downstream reading comprehension performance. Instead, they proposed a sliding window segmentation approach to create passages from Wikipedia articles. In line with this methodology, we used the same approach to create passages for our cross-lingual question-answering experiments.

To create our passages, we downloaded the Wikipedia dumps dated May 01, 2022, for English Wikipedia and April 20, 2022, for French Wikipedia. We then applied the sliding window approach to generate fixed-length passages of 100 tokens each from these dumps. These passages serve as our knowledge base for retrieval and answer span extraction. By adopting the sliding window segmentation approach for creating Wikipedia passages, we aim to improve downstream reading comprehension performance. The fixed-length passages enable efficient indexing and retrieval of relevant information for a given question while reducing the impact of disjoint and fragmented information that may occur when arbitrarily splitting articles.

## B  Training and Evaluation Details

### B.1  mDPR Reader:

We train a multilingual DPR reader model using pretrained bert-base-multilingual-uncased [9] as the model backbone. The model was trained to predict the correct answer span for a question given a set of relevant passages. We trained our model using the DPR retriever output[10] on the training and development set of Natural questions and evaluated on the test set of AFRIQA in a zero-shot manner. The model was trained on 4 A6000 Nvidia GPUs with a batch size of 16 and 2 gradient accumulation steps. We used an initial learning rate of 5e-5 and 500 warmup steps. The full list of training hyperparameters can be found in Table 9.

### B.2  AfroXLM-R Reader

To extract answer spans from the gold passages, we train extractive reader models on the training set of Squad 2.0 (Rajpurkar et al., 2016) and fQuad (d'Hoffschmidt et al., 2020) using AfroXLM-R as a backbone. We evaluated the models on the test queries and the annotated gold passages. The models were trained for 5 epochs using a fixed learning rate of 3e-5 and batch size of 16 on a single A100 Nvidia GPU.

### B.3  mT5 Reader

We finetuned multilingual pretrained text-to-text transformer (mT5) (Xue et al., 2020) on Squad 2.0 (Rajpurkar et al., 2016) dataset to generate answers from the gold passages. We trained the model for 5 epochs with a learning rate of 3e-5 and batch size of 32 on a single A100 Nvidia GPU.

## C  Machine Translation BLEU Scores

Table 10 shows the BLEU score of the different translation systems on the test set of AFRIQA, evaluated against the human-translated queries. Google Translate performs the best on the languages it supports while NLLB 1.3B achieves slightly poorer performance with a broader language coverage.

## D  Additional Experiments

### D.1  Retrieval Top-20/100 Accuracy

We present top-20 retriever accuracy results in Table 11.

---

[9] https://huggingface.co/bert-base-multilingual-uncased
[10] https://github.com/facebookresearch/DPR

| Source lang | Target lang | GMT | NLLB | M2M-100 |
|---|---|---|---|---|
| bem | eng | — | **24.4** | — |
| fon | fre | — | **16.6** | 8.7 |
| hau | eng | **55.2** | 44.6 | 26.3 |
| ibo | eng | **48.3** | 46.3 | 34.1 |
| kin | eng | **44.9** | 43.1 | — |
| swa | eng | **54.0** | 53.2 | 34.7 |
| twi | eng | **33.0** | 30.1 | 15.7 |
| wol | fre | — | **16.6** | 12.7 |
| yor | eng | **32.7** | 30.6 | 10.6 |
| zul | eng | **50.2** | 45.4 | 33.3 |
| avg | — | **45.5** | 35.1 | 22.0 |

Table 10: **Translation BLEU Scores:** BLEU score of some translation systems on the test set for the answer translation task. Note that Google Translate is not yet available in all languages, due to their very low-resource nature.

This further highlights the downstream effect of translation quality on retriever effectiveness with human translations showing better accuracy than other machine translation systems.

## D.2 XOR-Full Results

Table 12 presents the Exact Match Accuracy and BLEU scores of the XOR-Full task. The table contains downstream results of different translation-retriever pipelines to extract the answer span and translate it back to the same language as the question.

## E Summary of Language Linguistic Properties

In Table 13, we provide a structured breakdown of the typologies, grammatical structures, and phonology of the 10 languages in AFRIQA.

| | Human Translation | | | GMT | | NLLB | | M2M-100 | | Crosslingual |
| | BM25 | mDPR | Hybrid | BM25 | mDPR | BM25 | mDPR | BM25 | mDPR | mDPR |
|---|---|---|---|---|---|---|---|---|---|---|
| lang | | | | | **Recall@20** | | | | | |
| bem | 64.3 | 72.6 | **76.8** | — | — | 60.2 | 65.3 | — | — | 22.0 |
| fon | 71.5 | 72.2 | **74.6** | — | — | 49.6 | 52.3 | 46.5 | 46.9 | 30.3 |
| hau | 64.3 | 73.3 | **78.0** | 60.0 | 70.0 | 59.3 | 68.7 | 43.3 | 51.7 | 20.0 |
| igb | 75.3 | 78.7 | **87.8** | 72.4 | 76.0 | 70.2 | 73.4 | 67.2 | 74.3 | 34.0 |
| kin | 67.4 | 72.6 | **80.1** | 63.1 | 68.6 | 62.0 | 65.7 | — | — | 19.3 |
| swa | 54.6 | 67.6 | **72.5** | 52.7 | 66.9 | 50.3 | 64.6 | 47.0 | 61.3 | 26.8 |
| twi | 69.0 | 71.4 | **78.4** | 61.0 | 63.7 | 55.9 | 58.6 | 49.8 | 53.9 | 26.3 |
| wol | 68.6 | **73.1** | 72.2 | — | — | 42.8 | 43.7 | 41.0 | 40.4 | 18.0 |
| yor | 62.7 | 72.6 | **77.7** | 58.4 | 66.9 | 58.1 | 65.7 | 41.9 | 41.9 | 31.3 |
| zul | 68.6 | 76.6 | **83.7** | 66.5 | 71.7 | 62.2 | 69.2 | 53.2 | 64.9 | 18.2 |
| | | | | | **Recall@100** | | | | | |
| bem | 76.8 | 81.9 | **84.7** | — | — | 70.4 | **74.2** | — | — | 37.3 |
| fon | 78.8 | 79.3 | **80.1** | — | — | **60.3** | 59.3 | 59.6 | 59.3 | 46.9 |
| hau | 77.7 | 83.3 | **84.7** | 77.7 | **79.3** | 75.0 | 77.7 | 58.3 | 64.3 | 34.3 |
| igb | 87.0 | 89.7 | **94.6** | 85.6 | **87.5** | 84.8 | 83.9 | 82.4 | 83.4 | 50.1 |
| kin | 78.1 | 81.3 | **87.0** | 75.2 | **78.1** | 74.1 | 77.0 | — | — | 30.3 |
| swa | 70.9 | 80.5 | **82.1** | 68.1 | **79.8** | 68.2 | 77.2 | 64.2 | 76.2 | 40.1 |
| twi | 78.4 | 82.9 | **85.7** | 71.6 | **83.7** | 70.0 | 72.5 | 61.8 | 63.1 | 38.4 |
| wol | 82.6 | 82.6 | **84.7** | — | — | **56.0** | 55.1 | 57.2 | 53.6 | 31.1 |
| yor | 78.6 | 83.4 | **87.1** | 73.2 | **79.2** | 71.1 | 78.3 | 59.6 | 55.4 | 46.7 |
| zul | 86.2 | 86.2 | **91.1** | **83.1** | 72.0 | 77.0 | 80.6 | 71.1 | 74.8 | 28.9 |
| avg | 79.5 | 83.1 | **86.2** | 76.4 | **79.9** | 70.8 | 73.6 | 64.3 | 66.3 | 38.4 |

Table 11: **Retrieval recall@20/100**: This table presents the retrieval recall@20/100 results for different translation types on the test set of AFRIQA. This shows the percentage of the top 20/100 retrieved passages that contain the answer. Crosslingual retrieval skips the translation step

| Translation | | | | | | XOR-Full BLEU | | | | | | | Average |
| Query | Answer | Retrieval | | bem | fon | hau | ibo | kin | swa | twi | wol | yor | zul | **BLEU** |
|---|---|---|---|---|---|---|---|---|---|---|---|---|---|---|
| GMT | GMT | BM25 | | — | — | 19.4 | 28.2 | 21.1 | 16.0 | 11.7 | — | 13.8 | 18.1 | 18.3 |
| GMT | GMT | mDPR | | — | — | 20.1 | 30.3 | 23.3 | 19.9 | 13.2 | — | 18.6 | 19.6 | **20.7** |
| NLLB | NLLB | BM25 | | 11.4 | 1.7 | 15.9 | 24.8 | 16.8 | 16.9 | 16.6 | 2.9 | 10.9 | 10.7 | 12.9 |
| NLLB | NLLB | mDPR | | 10.9 | 3.3 | 17.0 | 27.2 | 18.8 | 18.3 | 17.5 | 3.1 | 15.3 | 11.4 | 14.3 |
| | | | | | | XOR-Full EM | | | | | | | | **EM** |
| GMT | GMT | BM25 | | — | — | 16.3 | 21.0 | 12.3 | 10.9 | 4.0 | — | 8.0 | 12.0 | 12.1 |
| GMT | GMT | mDPR | | — | — | 15.7 | 22.7 | 15.0 | 14.6 | 4.9 | — | 12.7 | 14.2 | **14.2** |
| NLLB | NLLB | BM25 | | 6.7 | 0.5 | 11.7 | 15.4 | 7.8 | 10.0 | 10.6 | 2.4 | 5.1 | 4.3 | 7.5 |
| NLLB | NLLB | mDPR | | 5.4 | 0.2 | 10.7 | 17.6 | 8.6 | 15.3 | 10.8 | 2.4 | 7.2 | 4.9 | 8.3 |

Table 12: XOR-Full results

| lang | Family | Tenses | Negation | Plurality | WH-questions |
|---|---|---|---|---|---|
| **bem** | Niger–Congo | Affix to head word present "ali", past "aali" | Affix to head word: "ta", "shi", and "kaana" | Affix to the steam of the word depending on noun class | What: "cinshi", Who: "naani" When: "iisa", Why: "mulandunshi" Which: "ciisa", Where: "kwi/kwiisa" |
| **fon** | Niger–Congo | New word added: past "xóxó" | New word added: "a" | New word added: "lɛ" | What:"Etɛ", Who: "Mɛ" When: "Hwetɛnu", Why: "Aniwú" Which: "ɖe tɛ", Where: "Fite" |
| **hau** | Afro–Asiatic | Indicative form Words used to indicate tenses: past: "tsohon" (was) present: "yanzu" (is) | Indicative form. Words used to indicate negation: ba/ba a" (not) and "banda" (except) | Suffix with vowel deletion. E.g.: "hula" (cap), "huluna" (caps) "mace" (girl), "mataye" (girls) | What: "mè/ya", Who: "wa" When: "yaushe", Why: "dan me/akan me" Which: "wanne", Where: "ina/ a ina" |
| **igb** | Niger–Congo | None | Suffix "ghi" | No suffix. Count is often specified after the word | What: kedu/gini, Who: onye/kedu onye When: kedu mgbe, Why: gini mere/gini Which: kedu nke, Where: ebee How: kedu ka or kedu etu |
| **kin** | Niger–Congo | Changes to morphemes in a word | Changes to morphemes in a word | Changes to morphemes in a word | What: "iki", Who: "nde/inde" When: "ryari", Which: "ikihe/uwuhe" Where: "hehe", How: "gute" |
| **swa** | Niger–Congo | Present: "ni" (is), Past: "alikuwa" (was/former) Future: "atakuwa" (will be) | — | Indicated by changes to the prefix according to noun class | What: "nii", Who: "nani", When: "lini" Why: "kwanini", Which: "upi", Where: "upi", How: "vipi" |
| **twi** | Niger–Congo | None | "n" is added to the root word | Indicated by replacing the first two letters of a root word with "mm" or "nn". | What: "ɛdeen", Who: "hwan", When: daben, Why: aden, Which: deɛhen Where: ɛhenfa, How: sɛn |
| **wol** | Niger–Congo | Dependent word: past tense, "oon" is attached to the end of the verb | Keyword "ul" is added at the end of the verb e.g nekk – ¿ nekkul | Dependent word: plurality, "yi" or "ay" is attached before or after the word | What: ian, Who: kan, When: kañ Why: lu tax, Which: ban, Where: fan, How: naka |
| **yor** | Niger–Congo | To indicate present tense, keyword "n". Past tense is indicated with "ti" with or without a time period | Keywords such as *kò. máa, nile* | Count is specified with a word | What: "Kini", Who: "Tani" When: "iga / nigba", Why: "kilode" Which: "ewo", Where: "Nibo" |
| **zul** | Niger–Congo | Present: affix after subject concord (e.g. "ya" or "sa") Past: suffix (e.g. "e" or "ile") | Typically indicated by the prefix "nga- | Indicated by morphemes "aba", "izi", "imi", "o" | What: "yini", Who: "ubani", When: "nini" Why: "kungani", Which: "yiliphi", Where: "kuphi", How: "kanjani" |

Table 13: **Language Linguistic Features:** This table provides a breakdown of the typologies, grammatical structures, and phonology of the 10 languages in AFRIQA