# OpenReview forum: "Cross-lingual Open-Retrieval Question Answering for African Languages"
_EMNLP/2023/Conference — EMNLP 2023 Findings_

### Official Review · Reviewer_aGgm · 2023-08-03

**Soundness:** 3

**Excitement:**

3: Ambivalent: It has merits (e.g., it reports state-of-the-art results, the idea is nice), but there are key weaknesses (e.g., it describes incremental work), and it can significantly benefit from another round of revision. However, I won't object to accepting it if my co-reviewers champion it.

**Paper Topic And Main Contributions:**

The manuscript introduces a new dataset for cross-language question answering focusing on African languages. The main contribution is the dataset, which includes 10 African languages plus two supporting languages, English and French.

**Questions For The Authors:**

Question A: It is unclear why the linguistic properties of each of the languages are crucial when formulating the questions.

Question B: I somehow miss an explanation on why precisely these 10 languages are included and not others.

Question C: Section 2.2. What is the exact financial compensation provided to each contributor?

Question D: Up to 40% of the questions in certain languages lack any question (at least 12%). Is this imbalance reasonable/representative? Could the authors justify why this large amount of unanswerable questions make the dataset realistic?

Question E: That google does not cover a certain language could simply mean that the company is not necessarily interested in it commercially. Therefore, I suggest to dim the claim in footnote 6.

Question F: The authors should make it clear what exactly the models are. Is BM25 the only monolingual one? Why not DPR beside mDPR? Thsi is confusing in Section 4.2 and in 5.1 as well.

Question G: Line 463. Could the authors give some context? What is the situation with other languages? Is it similar?

**Reasons To Accept:**

The authors have developed a good new resource and include reasonable experimentation.

**Reasons To Reject:**

Some of the details of the corpus construction, as well as of the experiments, are missing. Some decisions lack support.

**Reproducibility:**

3: Could reproduce the results with some difficulty. The settings of parameters are underspecified or subjectively determined; the training/evaluation data are not widely available.

**Reviewer Confidence:**

3: Pretty sure, but there's a chance I missed something. Although I have a good feel for this area in general, I did not carefully check the paper's details, e.g., the math, experimental design, or novelty.

**Typos Grammar Style And Presentation Improvements:**

The authors say "we demonstrate" (l. 88). They do not demonstrate. They show

Figure 1 should be larger. The font size should be at least that of the footnotes.

evaluate performance -> the performance

---

> ### Author Rebuttal · Authors · 2023-08-29
>
> We would like to thank the reviewer for their feedback! We have addressed the different points raised below:
>
> -  Linguistic Properties: We talk about the linguistic features of these languages to highlight their diversity. This is important because our work focuses on translating questions from one language to another and understanding these unique features helps in the translation process. That said, when it came to actually creating the questions in our dataset, we gave our annotators full freedom. We didn't enforce any specific linguistic guidelines. So, the linguistic properties are useful for the bigger picture but were not directly used in making the questions. I hope this clarifies our approach.
>
> - Selection of 10 languages: Thank you for pointing out the need for clarity on our language selection. The primary reason we focused on these 10 African languages is because we had access to native speakers who could effectively handle the tasks of creating, translating, and annotating questions in these languages. Finding annotators who are proficient speakers of many African languages can be a challenge, and it was pivotal for our research to have individuals with a good grasp of their respective languages. We will ensure that this is explicitly stated in the updated version of our manuscript.
>
> - Financial Compesation: The total budget for each team is $3600 including the team coordination stipend. However, as mentioned in Section 2.2 Data Collection Procedure, team size ranges from 2-6 native speakers.
>
> - Unanswerable  Questions:     To address this, it's crucial to first refer to our methodology for question generation: We asked our annotators to ask questions based on their genuine curiosity, rather than being restricted to specific passages. Afterward, we tried to find answers to these questions from Wikipedia, a common source used in well-known QA datasets like Natural Questions and TyDi QA, in English or French depending on the African language. The high rate of unanswerable questions, particularly for the 10 African languages in our collection, comes from Wikipedia's limited coverage of topics directly related to Africa. In the paper, we discuss this point specifically in lines 337-339.
>
> - Google Translate coverage:      We appreciate the reviewer's insight. Indeed, Google's decision not to cover certain languages might be tied to commercial interests, not just the low-resource status of these languages. We'll adjust the footnote accordingly.
>
> - Retrieval Models: Thank you for bringing this to our attention. In our study, we used three main retrieval methods:
>     - BM25: This is a keyword-based search method. We had separate indexes for both French and English texts making it a monolingual search.
>     - mDPR (multilingual Dense Retriever): Unlike the typical DPR that relies on an English-only BERT model, mDPR uses a multilingual BERT model. This was crucial for our experiments, which looked at both English and French texts and accepted questions in 10 different languages.
>     - Hybrid Retriever: This method combines the results from BM25 and mDPR to improve performance.
>   We opted for mDPR over the standard DPR specifically because mDPR can work well with multiple languages, aligning more closely with the goals of our research. In the revised version of our paper, we'll be sure to spell out these details more clearly to avoid any confusion.
>
> - Could the authors give some context? What is the situation with other languages? Is it similar?: Line 463 states that “cross-lingual representations learned by current retrieval methods are not yet of sufficient quality to enable accurate retrieval across different languages.” This refers to using a single multilingual model to perform retrieval across multiple languages. This is still an active area of research and even for high-resource languages, translation-based approaches (query/document translation) still yield better results compared to cross-lingual dense retrieval methods. I hope that gives a clearer picture of our point. Thanks for allowing us to explain further.

---

### Official Review · Reviewer_Fe2K · 2023-08-04

**Soundness:** 4

**Excitement:**

4: Strong: This paper deepens the understanding of some phenomenon or lowers the barriers to an existing research direction.

**Paper Topic And Main Contributions:**

The main contribution of this work is the creation of the first open-retrieval cross-lingual question answering dataset focusing on African languages. The provide collection procedure for this dataset, analysis on the features, as well as benchmarking on cross-lingual retrieval task. The paper is well-written and the dataset creation is well-motivated and supported.

**Questions For The Authors:**

- How would one expand this dataset for new languages?
- Have you done any quality checks to mitigate possible toxic cases?

**Reasons To Accept:**

- Valuable resource creation
- Good benchmarking
- Well-presented.

**Reasons To Reject:**

None

**Reproducibility:**

4: Could mostly reproduce the results, but there may be some variation because of sample variance or minor variations in their interpretation of the protocol or method.

**Reviewer Confidence:**

4: Quite sure. I tried to check the important points carefully. It's unlikely, though conceivable, that I missed something that should affect my ratings.

---

> ### Author Rebuttal · Authors · 2023-08-28
>
> We would like to thank the reviewer for their feedback!
> We have addressed the different points raised below:
>
> - How would one expand this dataset for new languages: In the final version of our paper, we'll include a link to a repository outlining our methodology which can be extended to create the dataset for new languages. This repository contains reproducible code for running baseline retrieval and reading comprehension experiments. We've also made our annotation tools available as open source, complete with comprehensive documentation. This will facilitate anyone interested in extending the dataset using the tools we developed. We removed the link in the current paper version due to anonymity.
>
> - Have you done any quality checks to mitigate possible toxic cases: Yes, In Section 2.3 Quality Control from line 317 we try to explain the quality control process. Annotators flagged inappropriate questions when translating or during answer annotation. Additionally, language coordinators further checked the question and answer pairs.

---

### Official Review · Reviewer_u1CQ · 2023-08-10

**Soundness:** 4

**Excitement:**

4: Strong: This paper deepens the understanding of some phenomenon or lowers the barriers to an existing research direction.

**Paper Topic And Main Contributions:**

The paper is about a new dataset focused on low resource languages for cross lingual retrieval and question answering. In this case it is focused on 10 African languages.
The authors uses and evaluated manual translations and automatic translations to generate the dataset.
Experiments were conducted using sota retrieval approaches as BM25, DPR and hybrid systems.

**Questions For The Authors:**

1 Can you better clarify how is quality of the translations monitored and evaluated?

**Reasons To Accept:**

+ new dataset for xor-qa and low resources languages (10 africans languages)
+ clear experiments and clear results
+ comparison of human translations and machine translations
+ large set of proposed experiments
+ dataset released with cc license

**Reasons To Reject:**

+ better paper structure (e.g: related work section need to be moved before experiments for a better clarification )
+ quality control section is not completely clear


**Reproducibility:**

4: Could mostly reproduce the results, but there may be some variation because of sample variance or minor variations in their interpretation of the protocol or method.

**Reviewer Confidence:**

5: Positive that my evaluation is correct. I read the paper very carefully and I am very familiar with related work.

---

> ### Author Rebuttal · Authors · 2023-08-28
>
> Thanks for your feedback! We are encouraged that you find our work beneficial to the study of low-resource languages.
> We have addressed the different points raised below:
>
> - Better Paper Structure:  Thank you for pointing out the issue with the paper structure. As suggested we will move the related work section before the experiments when we prepare the final version of the paper.
>
> - Monitoring and evaluating the quality of translations: In Section 2.3, "Quality Control", we detailed the multi-step process employed to ensure the quality of translations. Initially, annotators identified potential errors in both the local and pivot languages. These flagged translations were then reviewed by our language coordinators for accuracy in both the questions and answers. Finally, a central project manager further validated these translations for consistency across the dataset. It is worth noting that we did not aim for perfect translation but for naturalness to reflect how any speaker would pose a question in the language, hence why professional translators were not engaged in the translation phases.  We will make sure to improve the clarity of this section in the final submissions.

---

### Meta-Review · Area_Chair_VC7a · 2023-09-18

**Recommendation:** 4

**Metareview:**

The main contribution of this paper is a cross-lingual question answering/retrieval dataset for 10 African languages (+ English and French), generated through automatic and human translation. The dataset is claimed to be the first of its kind for these languages.

The reviewers agree that the corpus building corpus and the benchmarking and analyses are generally clearly presented. The resulting resource is expected to be very valuable. On the other hand, the reviewers found the structure of the paper a bit unintuitive and were missing some details. Also, some decisions should be better justified. However, these issues mostly concern the presentation of the paper and can be addressed in the camera-ready version.

---

### Decision · Program_Chairs · 2023-10-07

**Decision:**

Accept-Findings

**Comment:**

The main contribution of this paper is a cross-lingual question answering/retrieval dataset for 10 African languages (+ English and French), generated through automatic and human translation. The dataset is claimed to be the first of its kind for these languages.

The reviewers agree that the corpus building corpus and the benchmarking and analyses are generally clearly presented. The resulting resource is expected to be very valuable. On the other hand, the reviewers found the structure of the paper a bit unintuitive and were missing some details. Also, some decisions should be better justified. However, these issues mostly concern the presentation of the paper and can be addressed in the camera-ready version.